# Understanding the complementarities of surface-enhanced infrared and Raman spectroscopies in CO adsorption and electrochemical reduction

Xiaoxia Chang[1,2,3], Sudarshan Vijay[4], Yaran Zhao[3], Nicholas J. Oliveira[3], Karen Chan [ID] [4✉] & Bingjun Xu [ID] [1,2,3✉]

In situ/operando surface enhanced infrared and Raman spectroscopies are widely employed in electrocatalysis research to extract mechanistic information and establish structure-activity relations. However, these two spectroscopic techniques are more frequently employed in isolation than in combination, owing to the assumption that they provide largely overlapping information regarding reaction intermediates. Here we show that surface enhanced infrared and Raman spectroscopies tend to probe different subpopulations of adsorbates on weakly adsorbing surfaces while providing similar information on strongly binding surfaces by conducting both techniques on the same electrode surfaces, i.e., platinum, palladium, gold and oxide-derived copper, in tandem. Complementary density functional theory computations confirm that the infrared and Raman intensities do not necessarily track each other when carbon monoxide is adsorbed on different sites, given the lack of scaling between the derivatives of the dipole moment and the polarizability. Through a comparison of adsorbed carbon monoxide and water adsorption energies, we suggest that differences in the infrared vs. Raman responses amongst metal surfaces could stem from the competitive adsorption of water on weak binding metals. We further determined that only copper sites capable of adsorbing carbon monoxide in an atop configuration visible to the surface enhanced infrared spectroscopy are active in the electrochemical carbon monoxide reduction reaction.

[1] College of Chemistry and Molecular Engineering, Peking University, Beijing 100871, China. [2] Beijing National Laboratory for Molecular Sciences, Beijing 100871, China. [3] Center for Catalytic Science and Technology, Department of Chemical and Biomolecular Engineering, University of Delaware, Newark, DE 19716, USA. [4] CatTheory Center, Department of Physics, Technical University of Denmark, Kongens Lyngby 2800, Denmark. ✉email: kchan@fysik.dtu.dk; b_xu@pku.edu.cn

Electrosynthesis, including the electrochemical $CO_2$ and CO reduction reactions, referred to as the $CO_2RR$ and the CORR, respectively, has been recognized as a key component in the decarbonization of the energy and chemical sectors with the increasingly affordable and available renewable electricity[1–3]. A key enabling factor in the development of effective catalytic materials for electrosynthesis is the understanding of how active sites on the catalyst surface facilitate the bond breaking and forming processes in electrochemical transformations[4]. The complexity of electrified solid-liquid interfaces, at which most electrocatalytic transformations occur, renders many ex-situ and non-interfacial specific techniques less effective in providing insights relevant to the reaction conditions[5]. In this regard, surface-enhanced vibrational spectroscopies, including infrared (IR) and Raman, have emerged as potent tools to determine surface compositions[6–9], identify reaction intermediates[10–12], and elucidate mechanisms[13–15] for their excellent compatibility with in situ/operando studies.

Surface-enhanced IR and Raman spectroscopies referred to as SEIRAS and SERS, respectively, are complementary techniques in electrocatalytic investigations. It is well established that IR and Raman as molecular spectroscopies provide complementary information, derived from their sensitivity toward the dipole moment and polarizability, respectively, of the substrate. When employed for electrochemical interfacial studies, significant interfacial specificity is required for these techniques, since signals from bulk electrolytes could easily overwhelm the information about the interface. Although the main mechanism for the surface enhancement in SEIRAS and SERS is considered to be similar, i.e., enhanced surface plasma afforded by rough metal surfaces (an electromagnetic mechanism)[16–19], several fundamental and practical considerations make these two techniques complementary, more so than their counterparts in molecular spectroscopies.

Firstly, Raman spectroscopy is less sensitive to water than IR spectroscopy, which makes the Kretschmann configuration necessary for SEIRAS. This configuration takes advantage of the attenuated total reflection (ATR) mechanism to avoid excessive absorption of the IR beam in passing through a layer of aqueous electrolyte[20,21]. The optical property of common ATR crystals, as well as the surface chemistry needed for preparing robust metal films on the surface of the crystal, limits the lower bound of the spectral window in ATR-SEIRAS to ~1000 $cm^{-1}$. Thus, most M-O (M stands for a metal), M-C, and M-N vibrational modes are outside the spectral window of SEIRAS. Meanwhile, SERS' spectral window could be extended to as low as tens of wavenumbers, enabling it to investigate surface speciation. However, SEIRAS typically possesses better signal-to-noise ratios and higher temporal resolutions within its spectral window than SERS. Secondly, the surface enhancement effect with IR appears less metal-specific than Raman spectroscopy. While Cu, Ag, and Au are the three main SERS active metals, SEIRAS can be conducted on many more metallic surfaces, e.g., Pt, Pd, and Ni[17]. The development of the intensity borrowing strategy has extended the applicability of SERS to a wider variety of metal surfaces[22]. Thirdly, chemical enhancement has been well established in SERS, but documented with SEIRAS. Therefore, there could be varying degrees of discrimination towards surface species depending on these two different techniques.

Despite the aforementioned, well-known complementarities, SEIRAS and SERS have rarely been employed in concert in electrocatalytic studies. SEIRAS has been effective in identifying reaction intermediates in the $CO_2RR$ and CORR[11,14,23,24]. Surface speciation and reaction mechanisms have been successfully interrogated by SERS in the oxygen evolution and reduction reactions[12,25], as well as $CO_2RR$ and CORR[8,9,26,27], such as our

in situ observation of oxygen-containing species on Cu surfaces in CORR[8,9]. The scarcity of studies employing both types of surface-enhanced vibrational spectroscopies could be due to the implicit assumption that, for adsorbed species visible in both techniques, they would yield the same information. This assumption could originate from the well-known fact in molecular vibrational spectroscopies that, if a vibrational mode is both IR and Raman active, its vibrational bands in both spectroscopies will appear at the identical wavenumber. However, this assumption may not hold in the case of SEIRAS and SERS for two main reasons: (1) For an adsorbed species, e.g., CO, on different types of sites, e.g., terrace and defects, in various configurations could have different cross-sections in SEIRAS and SERS. Thus, adsorbates in a certain bonding environment could be discriminated or enhanced with a given technique, and the two techniques could be sampling different subpopulations of an adsorbate. (2) Difference in the chemical enhancement effect with SEIRAS and SERS is expected to introduce additional discrimination/enhancement to certain subpopulations of an adsorbate.

Therefore, there is a clear and urgent need to establish an effective methodology to use SEIRAS and SERS to obtain reliable structure-activity relationships in electrocatalysis. In the limited existing studies with both techniques, Weaver and coworkers reported significant differences in spectra of adsorbed CO on a roughened Au surface[28], along with good agreements in spectra of other adsorbed species, e.g., thiocyanide and azide, with SEIRAS and SERS. Such spectral discriminations of certain subpopulations of adsorbates could be induced by changes in (1) dipole moment and polarizability of the vibration normal mode; (2) effectiveness of the electromagnetic enhancement; and (3) effectiveness of the chemical enhancement. The first and third points are inevitably intertwined, as both originate from the electronic structure of the adsorbate and its immediate surroundings. This possibility raises a concerning point for the mechanistic studies of the $CO_2RR$ and CORR, in which adsorbed CO is often the only observable intermediate with vibrational spectroscopies, and its bands are frequently correlated with reactivities[11,15,29,30]. If only a subpopulation of the adsorbed CO is sampled by SEIRAS or SERS, there is no guarantee that the observed CO with one technique, as well as the corresponding adsorption sites, is responsible for the reaction activity. Thus, a better understanding of the limitation of using a single technique and the complementarity among different techniques in investigating complex electrified interfaces is a prerequisite for developing accurate predictive catalyst design principles for the CORR and beyond.

In this work, we investigated CO adsorption on Pt, Pd, Au and oxide-derived Cu (OD-Cu) surfaces using both SEIRAS and SERS under identical conditions in conjunction with density functional theory (DFT) computations. The same film of each metal was employed in these two spectroscopies to allow for direct comparisons. While CO bands on Pd observed by SEIRAS and SERS at the same conditions are quite consistent with each other in terms of peak position and Stark tuning rate, minor but reproducible differences in spectra from these two techniques are observed on the CO covered Pt surface. Substantial differences appear in SEIRA and SER spectra on relatively weak CO-binding surfaces, i.e., Au and OD-Cu. These observations suggest that the two surface-enhanced vibrational spectroscopies probe different subpopulations of adsorbed CO, e.g., CO adsorbed on surface sites with different local environments. Computational investigations with DFT indicate that $CO^*$ on different sites give rise to different IR and Raman intensities, due to differences in the derivatives of the adsorbate dipole moments and polarizabilities. A comparison of $CO^*$ and $H_2O^*$ binding energies on various metals also suggests that the difference in responses between

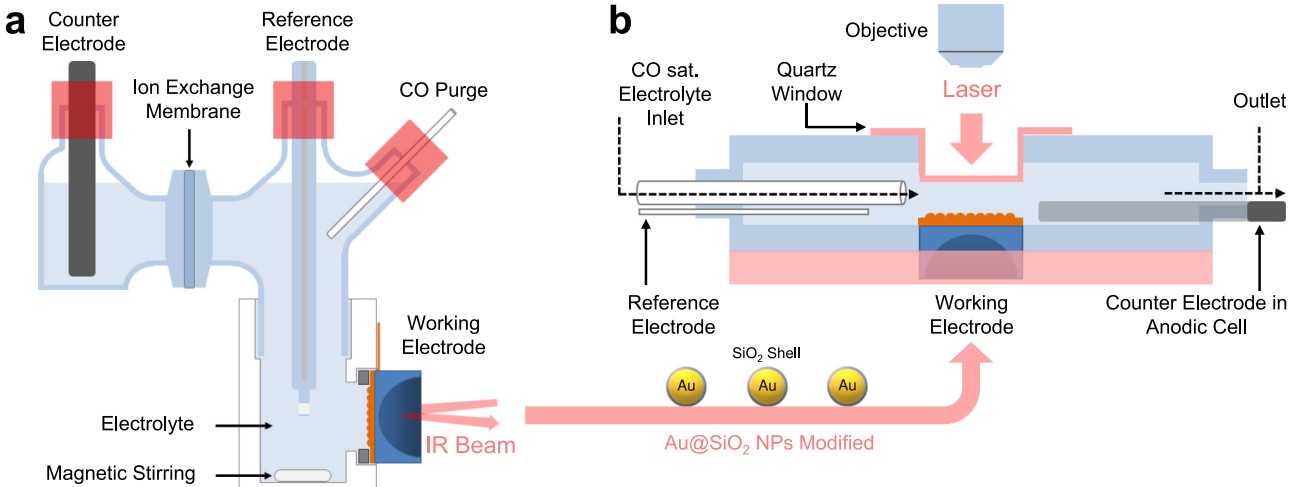

**Fig. 1 Schematic of the tandem in situ IR and Raman investigations. a** Schematic of the spectro-electrochemical cell with stirring function for in situ SEIRAS test. **b** Schematic of the flow cell for in situ SERS test. Both (**a**, **b**) contain two compartments that are separated by a piece of Nafion membrane.

different surfaces may stem from the competitive adsorption of water, which is more prominent on weak binding metals. Combining SEIRAS, SERS with real-time reactivity tracing, we show that only Cu sites corresponding to CO adsorbed in an atop configuration ($CO_{atop}$) visible in SEIRAS are active in the CORR.

## Results

**Tandem tests of in situ SEIRAS and SERS.** To enable direct comparisons between SEIRAS and SERS results, we developed an in situ spectroscopic cell and procedures to ensure the same surface was probed by both techniques. SEIRAS tests were conducted on a metal film deposited on a Si ATR crystal fitted in a home-designed spectro-electrochemical cell with a three-electrode configuration (Fig. 1a)[30]. After the SEIRAS experiment, the Si crystal was removed from the SEIRAS cell, and $SiO_2$-coated Au nanoparticles (Au@$SiO_2$) were introduced onto the film when necessary to enhance the Raman signal[22]. Au@$SiO_2$ particles have been shown to be chemically inert in several previous studies[8,22,31]. The Si crystal was then fitted into a custom-designed three-electrode SERS flow cell (Fig. 1b)[8,9]. The ability of both SEIRAS and SERS cells to house the Si crystal ensures the two techniques are probing the same surface, and thus results can be directly compared. A reference Si wafer was employed to calibrate the Raman shift by adjusting its lattice peak to $520.7\,cm^{-1}$ before each SERS test. In the following sections, CO is employed as a probe molecule to understand the difference between SEIRAS and SERS on strongly, moderately, and weakly adsorbing surfaces. Then, implications of the complementarity and difference between the two techniques in mechanistic studies are discussed using the CORR on OD-Cu as a model reaction.

**Tandem SEIRAS and SERS investigations of CO adsorbed on Pt and Pd surfaces.** The affinity of CO to Pt and Pd surfaces and its well-established adsorption behavior make them suitable model systems to compare IR and Raman results[32–36]. In this work, polycrystalline Pt films were directly deposited on the Si crystal using an electroless chemical plating method, while polycrystalline Pd films were deposited onto a gold substrate layer on the Si crystal through electrodeposition (detailed synthesis methods in Supplementary Information)[37,38]. The XRD patterns and SEM images of Pt and Pd films show the predominant (111) orientation and rough surface morphology (Supplementary

Fig. 1). The characteristic peaks of Au in the XRD patterns of Pd film originate from its Au substrate (Supplementary Fig. 1c). Both SEIRA and SER spectra on Pt and Pd were collected in CO saturated 0.1 M $HClO_4$ (pH 1.2).

On Pt, there are minor differences in the adsorbed CO bands between SEIRA and SER spectra. The primary band corresponding to CO adsorbed on the atop position of Pt, i.e., $CO_{atop}$, was observed in both SEIRA and SER spectra over the entire potential window investigated (0.6 to 0 V vs. reversible hydrogen electrode, or RHE), consistent with previous reports (Fig. 2a and Supplementary Fig. 2)[39,40]. All potentials reported in this paper are referenced to the RHE scale unless noted otherwise. The $CO_{atop}$ band in both SEIRAS and SERS redshifts as the potential becomes more negative, which is caused by the filling of the anti-bonding $2\pi^*$ orbital of CO and the vibrational Stark effect[41,42]. Interestingly, the measured Stark tuning rate is slightly lower with SERS ($21\,cm^{-1}$/V, green line in Fig. 2b) than with SEIRAS ($29\,cm^{-1}$/V, blue line in Fig. 2b). The Stark tuning rate determined with SEIRAS in this work is consistent with the previous reports under similar conditions[40,43,44]. Complementary Pt-C stretching mode was also observed in SER spectrum at lower wavenumbers with expected Stark shifts (Supplementary Fig. 3)[45]. It is worth noting that CO band intensity on Pt in both SEIRAS and SERS changes only slightly (less than 20 and 30%, respectively) in the investigated potential range of 0.6 to 0 V (Supplementary Fig. 2), indicating a negligible impact of CO dynamical dipole coupling on peak shift with the potential. In recent work, we showed that the impact of the coverage effect was less than 8% when the Stark tuning rate was determined using peaks with an integrated area greater than 60% of the maximum value in a given potential range[46]. Thus, the Stark tuning rates of CO on Pt in both SEIRA and SER spectra can be directly determined through the linear fitting of the band wavenumber vs. the applied potential (Fig. 1b). Different Stark tuning rates inevitably lead to different peak positions at different potentials, with the largest difference between SEIRAS and SERS being $4\,cm^{-1}$ at 0 V, which is within the spectral resolution employed in this work (Fig. 2a, b).

The sequence in which SEIRAS and SERS was conducted on the sample did not affect the CO bands, which was confirmed by the reproducible SEIRAS results on the same Pt film after SERS (Supplementary Fig. 2c). Current densities and cyclic voltammetry curves in SEIRAS and SERS tests on Pt are similar (Supplementary Fig. 4), indicating that the difference in the cell

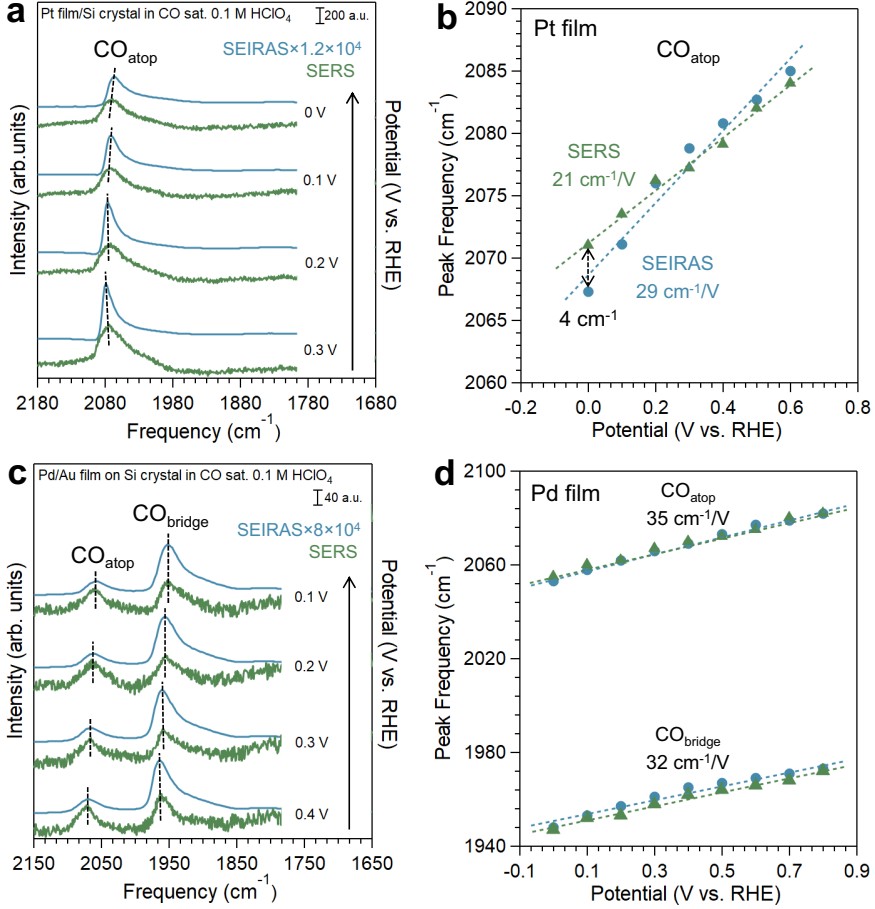

**Fig. 2 The comparison between tandem in situ SEIRA (blue) and SER (green) spectra on the same electrode.** The spectra on polycrystalline **a** Pt film and **c** Pd film in CO saturated 0.1 M HClO₄ (pH 1.2). The intensity of SEIRA spectra is multiplied by 12,000 and 80,000, respectively, in order to be plotted on the same scale as SER spectra. The dashed lines indicate the difference in peak frequencies between SEIRA and SER spectra under the same potential. The CO peak frequencies in SEIRA (blue circle) and SER (green triangle) spectra on **b** Pt film and **d** Pd film as a function of electrode potential. The Stark tuning rates are labeled.

configuration for these two spectroscopies does not impact the reactivity in any substantial way. Comparable current densities also entail similar interfacial pH, as well as the rates of H⁺ production or consumption. Thus, the different peak positions, and more importantly, Stark tuning rates of the $CO_{atop}$ band observed with SEIRAS and SERS suggest that the adsorbed CO species detected by these two techniques do not overlap entirely. A likely cause is that a subset of adsorbed $CO_{atop}$ in a specific configuration or on a specific type of microenvironment is selectively enhanced by one type of spectroscopy.

Intriguingly, CO bands on Pd observed by SEIRAS and SERS at the same condition are quite consistent with each other. Two sets of bands corresponding to CO adsorbed in the atop and bridge configurations ($CO_{atop}$ and $CO_{bridge}$, respectively) are present on the Pd surfaces in both SEIRA and SER spectra (Fig. 2c). Both bands redshift as the potential decreases (Supplementary Fig. 5), as expected from the vibrational Stark effect. The peak position shifts of $CO_{atop}$ and $CO_{bridge}$ bands with SEIRAS, 2082–2049 cm⁻¹ and 1973–1940 cm⁻¹, respectively, in the potential window of 0.8 to −0.2 V (Supplementary Fig. 5a) are consistent with previous reports[47,48]. The most notable feature of CO bands on Pd is that they are remarkably close in SEIRA and SER spectra collected at the same potentials (Fig. 2c and Supplementary Fig. 5), with the difference in peak position less than 3 cm⁻¹. It follows that the Stark tuning rates determined by SEIRAS and SERS are also similar for both

$CO_{atop}$ and $CO_{bridge}$ bands (Fig. 2d). Similar to the case on Pt, CO band intensities in SEIRAS and SERS largely remain constant within the potential range of 0.8 to 0 V (Supplementary Fig. 5), alleviating the need to remove the CO coverage effect on the Stark tuning rate (Fig. 2d). These results suggest that both techniques likely probe the same population of the adsorbed CO. It is important to note that the ratio between the integrated area of $CO_{atop}$ and $CO_{bridge}$ bands is closer to unity in SERS (0.98 at 0 V) than in SEIRAS (0.19 at 0 V, Supplementary Fig. 5). This suggests different enhancement effects of the same adsorbates with different adsorption configurations.

**Tandem SEIRAS and SERS investigations of adsorbed CO on Au surface.** Au as a well-known weakly adsorbing surface for CO provides another good limiting case in probing the difference between SEIRAS and SERS[11,35]. A chemically deposited Au film on a Si crystal (Supplementary Fig. 6) was used as the working electrode in CO saturated 0.1 M KHCO₃ (pH 8.9)[11]. The close to neutral pH of the electrolyte makes chemical erosion of the Au@SiO₂ during the SERS test unlikely. A single $CO_{atop}$ peak centered at 2118–2069 cm⁻¹ appears between 0.8 to 0 V, reaching a maximum peak intensity at ~0.4 to 0.5 V in SEIRAS (Supplementary Fig. 7a, b), similar to the results in a previous report[11]. Only a $CO_{atop}$ band is observed in SERS (Supplementary Fig. 8a), however, there are three substantial differences in the spectra between these two techniques: (1) The potential window within

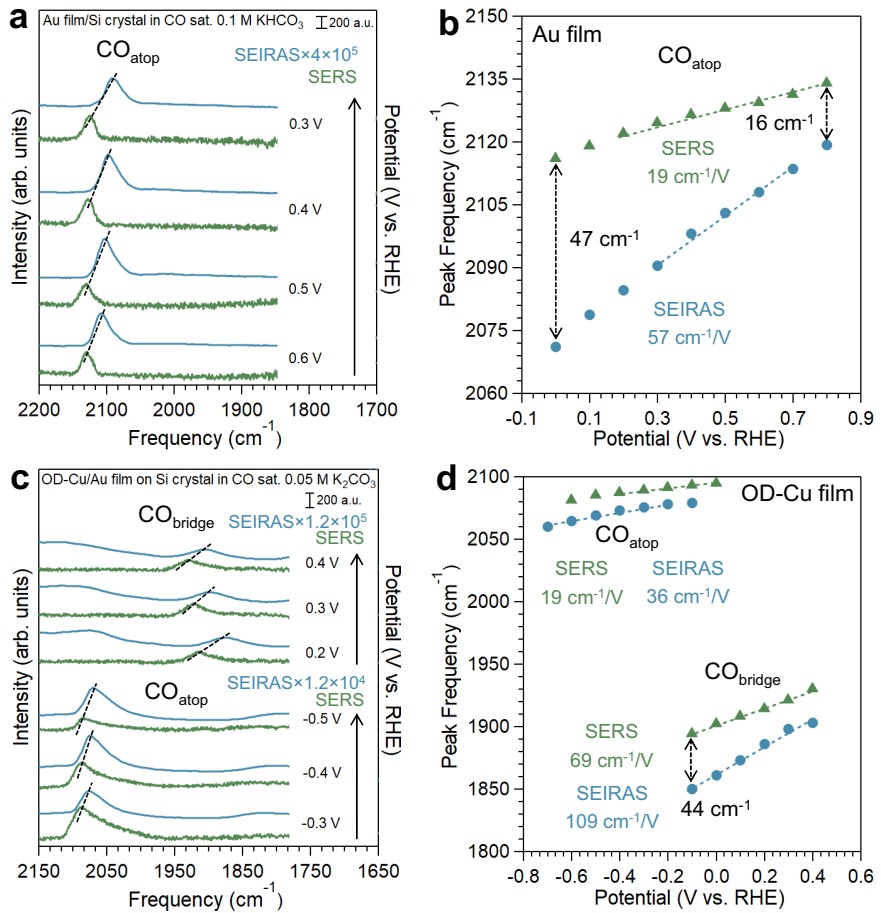

**Fig. 3 The comparison between tandem in situ SEIRA (blue) and SER (green) spectra.** The spectra on **a** the same polycrystalline Au film and **c** two fresh OD-Cu films in CO saturated 0.1 M KHCO$_3$ (pH 8.9) and 0.05 M K$_2$CO$_3$ (pH 10.6), respectively. The intensity of SEIRAS is scaled up in order to be plotted on the same scale as SERS. The dashed lines indicate the difference in peak frequencies between SEIRA and SER spectra under the same potential. The CO peak frequencies in SEIRA (blue) and SER (green) spectra on **b** Au film and **d** OD-Cu film as a function of electrode potential. The Stark tuning rates are determined by dashed lines and labeled in the figure.

which CO$_{atop}$ appears and the potential at which the CO$_{atop}$ band reaches its maximum are different (Supplementary Figs. 7a, 8a). While the CO$_{atop}$ band becomes barely visible at 0 V with SEIRAS, it reaches its maximum intensity at this potential in SER spectra. (2) Stark tuning rates of the CO$_{atop}$ band are substantially different (Fig. 3b). We found a Stark tuning rate of 19 cm$^{-1}$/V with SERS, and 57 cm$^{-1}$/V with SEIRAS. The CO surface coverage on Au changes substantially with the potential as evidenced by the varying peak area vs. potential in both SEIRAS (Supplementary Fig. 7a, b) and SERS (Supplementary Fig. 8a, b). As a result, the dynamical dipole coupling of CO, i.e., coverage effect, is expected to have a strong impact on the shift of peak position, which complicates the determination of the Stark tuning rate[49,50]. In order to remove the coverage effect, the Stark tuning rate was determined in the potential ranges of 0.7 to 0.3 V and 0.8 to 0.2 V with SEIRAS and SERS (Fig. 3b and Supplementary Figs. 7c, 8c), respectively, where all peak areas are greater than 60% of the maximum (Supplementary Figs. 7b, 8b) and the impact of coverage effect was less than 8%[45]. (3) The peak position of the CO$_{atop}$ band is consistently lower in SEIRAS than SERS with a maximum difference of 47 cm$^{-1}$ at 0 V, though the difference narrows as the potential increases (Fig. 3a, b). The distinct peak positions and Stark tuning rates of the bands suggest that subpopulations of CO$_{atop}$ in different local environments, e.g., CO adsorbed on terrace versus step sites on different facets, are sampled by the two spectroscopies. It is worth noting that Raman

spectra collected at different spots of the electrode may vary slightly as we reported on Cu microparticles recently[9]. However, variations in peak position among different spots are typically within 10 cm$^{-1}$ and cannot account for the substantial differences (up to 47 cm$^{-1}$) between IR and Raman spectra (Fig. 3). The magnitude of the difference in the peak positions and Stark tuning rate is larger on Au than Pt, indicating that the discrepancy between the two techniques may correlate with the adsorption strength of CO. As we discuss in the computational section, we postulate that this difference could arise from the competitive adsorption of water on weak binding metals and the resultant change in the distribution of CO as a function of potential.

**Tandem SEIRAS and SERS investigations of CO adsorbed on oxide-derived Cu.** The ability of Cu to selectively convert CO$_2$ into valuable multi-carbon products has been a focus of recent research[51]. Adsorbed CO is a known intermediate in the electrochemical reduction of CO$_2$ to multi-carbon products, and the moderate adsorption energy of CO on Cu has been proposed to be a key reason for its ability to facilitate C-C coupling reactions[35,52,53]. In particular, Cu surfaces after the oxidation-reduction treatment, referred to as oxide-derived Cu or OD-Cu, could significantly reduce the overpotential necessary for multi-carbon products[54,55]. Thus, OD-Cu is employed in the spectroscopic investigations for CO adsorption, which will be correlated

with reactivity later in this study. OD-Cu films were prepared by the reduction of $Cu_2O$ that was pre-electrodeposited onto the gold substrate layer on a Si crystal[14]. The OD-Cu film exhibits a predominant (111) orientation in the XRD pattern (Supplementary Fig. 9a) and roughened morphology as shown by the SEM image (Supplementary Fig. 9b). In contrast to the other metals investigated in this work (Pt, Pd, and Au), which are stable in the potential range investigated even after repeated potential scans, the Cu surface is known to reconstruct at negative potentials, especially in alkaline electrolytes. This instability leads to the varied intensity and lineshape of CO bands depending on the duration the Cu surface has been exposed to the negative potential[6,9,29,56,57]. Thus, to ensure SEIRA and SER spectra on OD-Cu were collected at comparable conditions, fresh OD-Cu electrodes were employed in SEIRAS and SERS tests, rather than using the same film in the two experiments in tandem. No $Au@SiO_2$ particles were introduced to the OD-Cu, as this surface is sufficiently rough to enable SERS[8,9].

SEIRA and SER spectra were collected during a cathodic potential step from 0.4 to −0.8 V in CO saturated 0.05 M $K_2CO_3$ (pH 10.6), followed by a reverse anodic scan back to 0.4 V. Both $CO_{atop}$ and $CO_{bridge}$ bands appear in SEIRA and SER spectra, however, in different potential windows and at slightly different wavenumbers. During the initial cathodic potential steps from 0.4 to −0.8 V, a weak $CO_{bridge}$ band at 1903–1778 $cm^{-1}$ appears in SEIRA spectra with the expected redshift due to the vibrational Stark effect (Supplementary Fig. 10a). Interestingly, no discernable $CO_{bridge}$ band shows up in the SER spectra under otherwise identical conditions (Supplementary Fig. 11a). One possibility is that the lower signal-to-noise ratio, or surface enhancement, of SERS is insufficient to detect such species. We consider this unlikely because a $CO_{bridge}$ band appears in the anodic scan in the SER spectra within the −0.1 to 0.4 V potential window (Fig. 3c and Supplementary Fig. 11a). It stands to reason that if the same bridge-bonded CO species is present in the initial cathodic scan at comparable coverages, the $CO_{bridge}$ band should be detected as well. In contrast, the $CO_{bridge}$ band is quite reversible with SEIRAS (Supplementary Fig. 10a). The peak position and Stark tuning rate of the $CO_{bridge}$ bands in SEIRA and SER spectra during the reverse anodic scan, where the $CO_{bridge}$ band is present in both spectra, do not agree with each other. The $CO_{bridge}$ bands have lower peak positions in SEIRAS than SERS, with a maximum difference of 44 $cm^{-1}$ at −0.1 V, and SEIRAS shows a larger Stark tuning rate (109 $cm^{-1}$/V) than SERS (69 $cm^{-1}$/V) (Fig. 3c, d). The Stark tuning rates of $CO_{bridge}$ bands were directly determined through linearly fitting the band wavenumber vs. the applied potential since the band intensities barely changed with the potential (Supplementary Figs. 10a, 11a). Similar observations were found with the $CO_{atop}$ band, which appears largely at the same potential (~0 V) during the initial cathodic steps (Supplementary Figs. 10a, 11a). The intensity of the $CO_{atop}$ band with SEIRAS peaks at −0.4 V, and the band is highly reversible (Supplementary Fig. 10a, b). In contrast, the $CO_{atop}$ band with SERS reaches its maximum intensity at −0.3 V and does not reappear at the subsequent anodic scan (Supplementary Fig. 11a, b). Different Stark tuning rates of 36 and 19 $cm^{-1}$/V are determined with SEIRAS and SERS, respectively (Fig. 3d). Similar to the case on Au, we determined the Stark tuning rate of $CO_{atop}$ band on OD-Cu using the peaks with integrated areas greater than 60% of the maximum, i.e., from −0.2 to −0.7 V with SEIRAS and 0 to −0.4 V with SERS (Fig. 3d and Supplementary Figs. 10c, 11c), to remove the coverage effect[45]. Based on these observations, the two spectroscopies are likely probing different subpopulations of both atop- and bridge-bonded CO. Peak positions and Stark tuning rates of $CO^*$ by SEIRAS and SERS on different electrodes employed were summarized in Table 1.

**Computation of IR and Raman intensities on transition metal surfaces and the impact of competitive water adsorption on potential-dependent vibrational frequencies.** To understand the differences in the spectral feature described above, we computed the IR and Raman intensities with DFT calculations for $CO^*$ on various sites and facets. The intensities were determined from dipole and polarizability derivatives, respectively, as calculated with a finite-difference grid and the eigenmodes of the dynamical matrix (see Computational Methods). We furthermore determined the $CO^*$ and $H_2O^*$ binding strengths from published temperature-programmed desorption (TPD) data. In what follows, we discuss the computed Stark tuning rates, the comparison between IR and Raman intensities, and the role of the competitive adsorption of water in the difference in IR and Raman responses with respect to potential in the different metals considered in this work.

Figure 4a–d shows the change in the computed $CO^*$ stretch frequency with potential for the top, bridge, and fcc sites on (111) terrace and (211) and (310) steps (see Supplementary Fig. 12 for schematics of the different sites) on Au, Cu, Pd, and Pt. As in the previous studies[58–62], we find the frequencies to be within the range 1900–2100 $cm^{-1}$ for top sites and lower than 1850 $cm^{-1}$ for bridge and fcc sites. These results are consistent with the theory of CO adsorption on metallic surfaces, which shows the adsorption strength to be determined by the backdonation to the $2\pi^*$ orbital of adsorbed CO[42,63]. The higher the coordination number of adsorbed CO has, the weaker the C-O bond is[64,65], leading to a lower wavenumber for the C-O stretching mode. Like in the experiments, we determined the Stark tuning rates from the computed potential-dependent CO stretching frequencies as follows:

$$\text{Stark tuning rate} = \frac{d\nu}{d(\text{potential})} \quad (1)$$

where $\nu$ is the C-O stretch frequency. Figure 4e illustrates the variation of the computed Stark tuning rates against the dipole moments of Au, Cu, Pd, and Pt surfaces. We find that the computed Stark tuning rates are in the range of 0–20 $cm^{-1}$/V, in line with the previous work[66,67], while the experimental values in this work are significantly higher in some cases (up to 100 $cm^{-1}$/V). The source of this apparently lower computed Stark tuning rate will be discussed below. Stark tuning rates and dipole moments generally track each other, i.e., a larger surface dipole moment of adsorbed CO gives rise to a larger corresponding change in dipole moment with potential. In particular, adsorbed CO on Au has the largest dipole moments (Fig. 4e). There is also a slight variation in the dipoles and corresponding Stark tuning rates on different facets for a particular type of site, but the variations in Stark tuning rate with the type of site tend to have a larger effect, similar to the variations in the CO binding strength.

In general, we find that computed IR and Raman intensities of adsorbed CO do not track each other at any given potential. Figure 4f to i shows the calculated IR and Raman intensities for adsorbed CO on all sites considered in Fig. 4a–d. There is no clear correlation between the IR and Raman intensities at any of the studied potentials (denoted by the color bar), which arises from the general lack of scaling between the derivatives of the dipole moments and polarizabilities. This general lack of scaling holds in the case of individual surface facets (different makers in Fig. 4f–i), though stepped surfaces tend to have comparatively larger IR intensities. This result supports the hypothesis that SEIRAS and SERS can probe different adsorbate subpopulations at a given interface, e.g., CO adsorbed on terrace versus step sites on different facets, and that the difference has its origins in the electronic structure of the adsorbate-metal system.

**Table 1 The peak positions and Stark tuning rates of CO* on Pt, Pd, Au, and OD-Cu, respectively.**

| | Pt | Pd | | Au | OD-Cu | |
|---|---|---|---|---|---|---|
| | $CO_{atop}$ | $CO_{atop}$ | $CO_{bridge}$ | $CO_{atop}$ | $CO_{atop}$ | $CO_{bridge}$ |
| SEIRAS | 2084−2067 cm$^{-1}$ | 2082−2053 cm$^{-1}$ | 1973−1948 cm$^{-1}$ | 2114−2090 cm$^{-1}$ | 2078−2060 cm$^{-1}$ | 1850-1903 cm$^{-1}$ |
| | (0.6 to 0 V) | (0.8 to 0 V) | (0.8 to 0 V) | (0.7 to 0.3 V) | (−0.2 to −0.7 V) | (−0.1 to 0.4 V) |
| | 29 cm$^{-1}$/V | 35 cm$^{-1}$/V | 32 cm$^{-1}$/V | 57 cm$^{-1}$/V | 36 cm$^{-1}$/V | 109 cm$^{-1}$/V |
| SERS | 2084−2071 cm$^{-1}$ | 2082−2055 cm$^{-1}$ | 1972−1947 cm$^{-1}$ | 2134−2122 cm$^{-1}$ | 2095−2087 cm$^{-1}$ | 1894-1930 cm$^{-1}$ |
| | (0.6 to 0 V) | (0.8 to 0 V) | (0.8 to 0 V) | (0.8 to 0.2 V) | (0 to −0.4 V) | (−0.1 to 0.4 V) |
| | 21 cm$^{-1}$/V | 35 cm$^{-1}$/V | 32 cm$^{-1}$/V | 19 cm$^{-1}$/V | 19 cm$^{-1}$/V | 69 cm$^{-1}$/V |

All the potentials are on the RHE scale. The potential ranges in which CO peak intensities are greater than 60% of the maximum are selected to determine the Stark tuning rates.

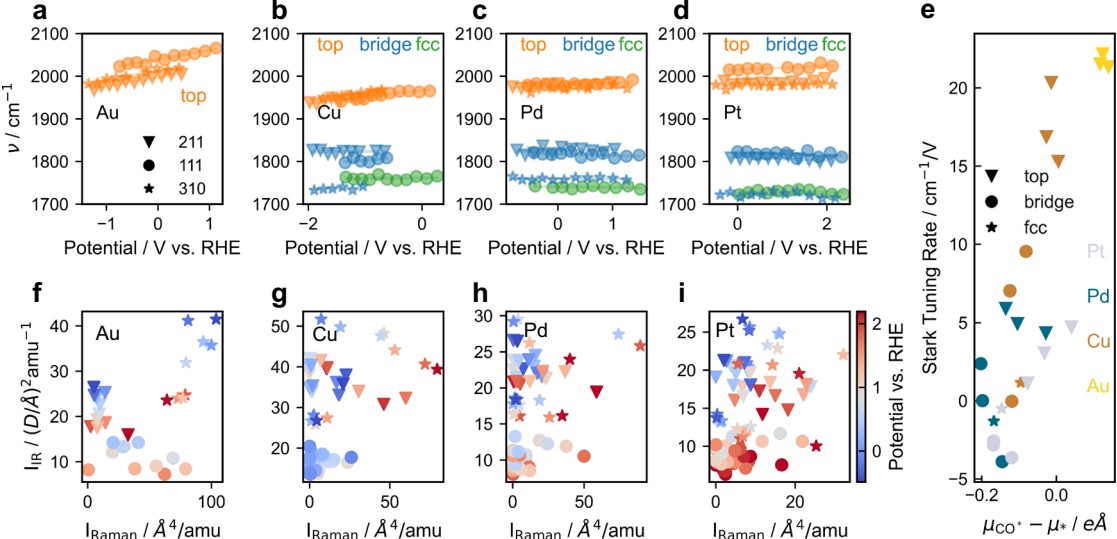

**Fig. 4 Computational investigations of CO* on various sites and facets of different metals.** DFT-computed vibrational frequencies as a function of the potential for **a** Au, **b** Cu, **c** Pd, and **d** Pt on top, bridge, and fcc sites on the 211, 111, and 310 facets. **e** Surface dipole (x-axis) and Stark tuning rates (y-axis) for all of the studied metals. Scatter plot of IR vs Raman intensities for **f** Au, **g** Cu, **h** Pd, and **i** Pt on facets and sites considered.

We now turn to the experimental observations that Cu and Au have different SERS/SEIRAS responses, while Pt and Pd do not. We suggest that this difference might arise from the degree of competitive water adsorption, which is more prominent in the weak binding metals of Cu and Au. In an aqueous environment, water can potentially compete for surface sites[68], depending on its affinity to the surface. On weak binding metals like Au, water and CO have similar binding strengths[69], which leads to their effectively weak adsorption on the surface. Figure 5a highlights this effect by showing the computed Boltzmann CO coverage as a function of different CO and $H_2O$ free energies of adsorption. The Boltzmann coverage of CO is given by the Eq. (2):

$$\theta_{CO^*} = \frac{\exp\frac{-\Delta G_{CO}}{k_B T}}{\sum_{i=CO^*,H_2O*,*}\exp\frac{-\Delta G_i}{k_B T}} \quad (2)$$

When $\Delta G_{CO} \sim \Delta G_{H_2O}$, the Boltzmann coverage of CO is between 1 ML (in red) and no perceptible coverage (in blue). We determined the free energies of adsorbed CO and $H_2O$ of the few representative transition metal facets in Fig. 5a directly from TPD experiments using the methodology in the literature[70]. Briefly, the desorption energies from a TPD curve are fitted to an expression containing the dilute coverage adsorption energy, adsorbate–adsorbate interactions, and the effect of configurational entropy (see Computational Methods). We use energies directly from TPD curves as opposed to DFT calculations in this analysis due to the current inability of static DFT calculations to

accurately predict water adsorption on transition metal surfaces[71–74].

In the case of Au(310), both adsorbed CO and $H_2O$ bind weakly ($\Delta G_{CO} \sim \Delta G_{H_2O}$). With comparable binding strengths, their coverages are lowered from what they would be in the absence of the other species. In contrast, on strong binding metals like Pt(111), Ni(111), and Rh(111), $\Delta G_{CO}$ is significantly larger than $\Delta G_{H_2O}$, and the interfacial water is unlikely to alter the CO coverage. Thus, the coverage of adsorbed CO at a particular site is determined by the relative binding energy of CO and $H_2O$, i.e., the stronger the CO binding strength against that of $H_2O$, the more likely that there would be a higher CO* coverage on that site (red region in Fig. 5a). Furthermore, competitive water adsorption could depend on the applied potential, since the water adsorption itself has been shown to depend on potential[75]. Recent ab initio molecular dynamics simulations showed the coverage of water increases from 0 ML to up to 0.5 ML on Pt(111) with an increase in the potential of 2 V, along with a concomitant increase in interfacial water-oriented with the oxygen end down towards the surface[75]. The same effect of decreasing water binding at reducing potentials was also observed on Cu(100)[76]. Meanwhile, we do not expect a significant change in CO binding strength with potential. Previous DFT calculations in the presence of an interfacial field suggest a change in CO binding strength of about 0.1 eV over the 0.5 V range[69,77].

We now apply this reasoning to our experiments, which show Au to have a Stark tuning rate of 19 cm$^{-1}$/V from SERS, and

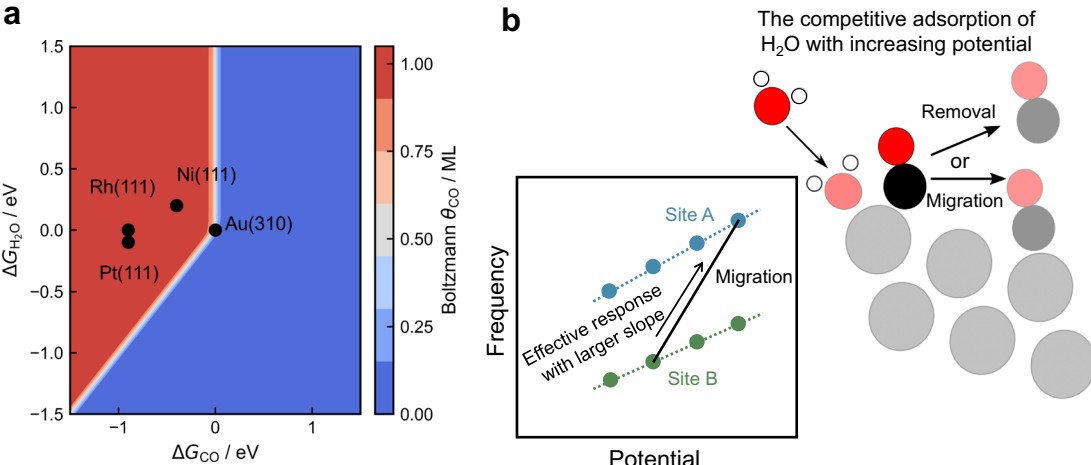

**Fig. 5 The competitive adsorption between CO* and H₂O*. a** Boltzmann coverages based on different $\Delta G_{CO}$ and $\Delta G_{H_2O}$ values with experimental data shown for Au(310)[92], Pt(111)[93,94], Ni(111)[95,96], and Rh(111)[97,98]. **b** Schematics showing the effect of the change in CO* distribution between Site B (green circles) and Site A (blue circles) on observed vibrational frequencies as a function of potential (left), and cartoon illustrating the effect of competitive H₂O adsorption on CO* binding site (right). Circles represent different atoms: hydrogen (white), carbon (black), and oxygen (red). The deep and light color shades indicate the molecules at lower and higher potentials, respectively.

57 cm⁻¹/V from SEIRAS (Fig. 3b). The computed Stark tuning rates (Fig. 4a) are ~20 cm⁻¹/V on all sites. Figure 5b (right panel) shows schematically a possible mechanism through which the electrode potential could impact the apparent Stark tuning rate based on our hypothesis. As the potential increases on the Au surface from 0.3 to 0.8 V$_{RHE}$, water adsorption has been calculated to be more favorable[75,76,78], which would give rise to a decreasing CO coverage on steps[69]. Meanwhile, the coverage of CO on terrace sites, which are less affected by water adsorption, is expected to either increase or remain constant based on the calculated adsorption energies of CO[69,79]. The large apparent Stark tuning rate would result from the gradual shift in the predominance of the signal of adsorbed CO on steps at lower potentials to that of adsorbed CO on terraces at higher potentials (see schematic in Fig. 5b's left panel), as CO adsorbed on terrace sites have a higher wavenumber than those on step sites based on the computational results (Fig. 4a). We note that a larger apparent Stark tuning rate would also appear if the coverage of CO on steps and terraces decreases with the decrease more significant on stepped sites. This effect is more pronounced in the IR than in Raman spectra because the calculated changes in the dipole moment derivative in adsorbed CO are more sensitive to a shift from the highly stepped Au(310) facet to Au(111) than that of the polarizability derivative (such as in Fig. 4f for Au), which qualitatively explains the discrepancy of the measured Stark tuning rates with the two techniques. This effect could also occur on Cu, which binds both CO and water weakly, however, the predominant adsorption sites could also change from the formation of an oxide phase at more anodic potentials. In contrast, in the case of Pt and Pd, coverages on the various sites are unlikely to change due to the dependence of water adsorption strength on potential. CO binds much more strongly than water on these surfaces, which suggests a fixed CO site distribution where the largest IR and Raman intensities dominate.

**Correlating spectroscopic and reactivity observations.** The contrasting results of the CO* bands on OD-Cu, Au, and to a lesser extent on Pt, in SEIRA and SER spectra under otherwise identical conditions point to a methodological dilemma in correlating spectroscopic results with reactivity data in electrocatalysis in general and the CORR in particular. Adsorbed CO is

not only a reaction intermediate in the CO₂RR and CORR, but also serves as a probe to identify various types of active Cu sites. The observations that the two surface-enhanced spectroscopies probe different subpopulations of the adsorbed CO raise the inconvenient possibility that CO adsorbed on the true active sites, even if it exists in sufficient coverage, could be overlooked with one or both techniques. To gain reliable structure-activity relations, it is clearly advantageous to employ multiple in situ techniques, e.g., SEIRAS and SERS, to obtain a comprehensive picture of the types of sites present at the reaction conditions, and reduce the likelihood of missing the adsorbed CO (or other probe molecules) on the true active sites. It is equally important to establish direct correlations between reactivity and spectral signatures, which would enable the identification of specific surface sites responsible for the observed reactivity.

One effective way of correlating spectral features with reactivity is to introduce a perturbation with known effects on the reactivity in spectroscopic experiments. Many recent works, including ours, have shown that the removal of forced convection, e.g., stirring, in a batch cell would significantly suppress the CORR by introducing the mass transport limitation of the scantly soluble CO[1,52,80]. The overall current density and the Faradaic efficiency for the CORR products drop precipitously when conducting the reaction without stirring in a batch cell in comparison with results obtained with stirring in both 0.05 M K₂CO₃ and 0.1 M KOH (Supplementary Fig. 13). It can be inferred that CO adsorbed on Cu sites active in the CORR would be preferentially consumed when the reaction is limited by the mass transport of CO. Both CO$_{atop}$ and CO$_{bridge}$ bands were monitored during the CORR at −0.7 V in 0.05 M K₂CO₃ before and after stopping stirring (Fig. 6). Right after the forced convection was removed, the intensity of the CO$_{atop}$ band with SEIRAS dropped with the overall current and recovered when the stirring resumed (Fig. 6b). Meanwhile, the CO$_{bridge}$ band remained largely unchanged throughout this period (Fig. 6a). This is a clear indication that the sites corresponding to the CO$_{atop}$ band are active in the reaction, while those corresponding to the CO$_{bridge}$ band are spectators or poisoned, as claimed in several recent publications[29,81]. The irreversibility of the CO$_{atop}$ band with potential steps in SERS (Supplementary Fig. 11a) suggests that the corresponding sites are unlikely to be the active sites in the CORR at the steady-state due to its structural instability. The possibility of some sites being so active at −0.7 V that adsorbed CO

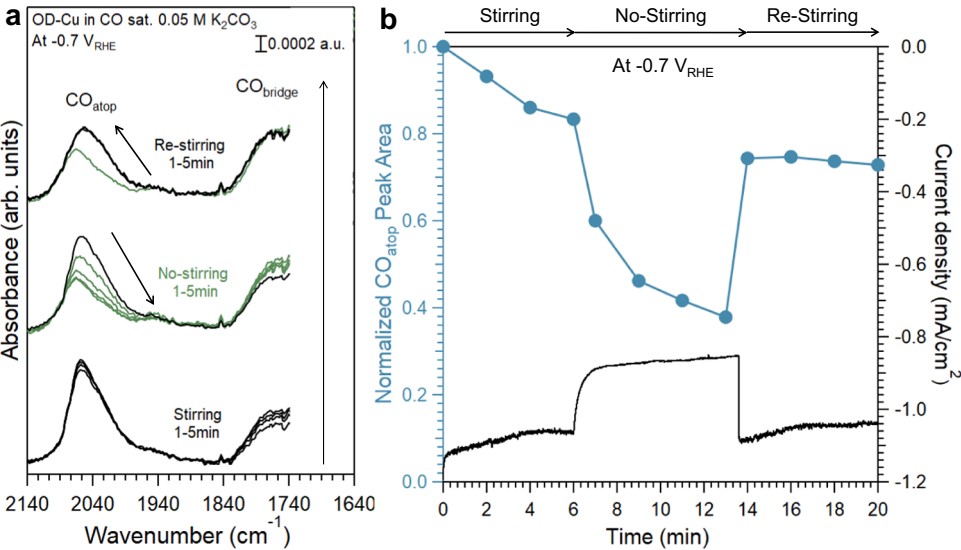

**Fig. 6 The impact of stirring on IR spectra and current density. a** In situ SEIRA spectra on polycrystalline OD-Cu film at −0.7 V in CO saturated 0.05 M K$_2$CO$_3$ (pH 10.6) with and without stirring. **b** The normalized peak area of CO$_{atop}$ bands (blue curve) and the current density (black curve) during the SEIRAS test in **a**.

has a residence time too short to be detected by SEIRAS or SERS (unless it is inactive in both spectroscopies) can also be reasonably ruled out. This is because any active sites capable of effectively converting CO are expected to have reasonable adsorption energy of CO, and it follows that such sites should adsorb CO when the potential is insufficiently negative to drive the CORR. No distinct and reversible CO$_{atop}$ band was observed at less negative potentials with SERS, suggesting that such sites do not exist in sufficient coverage to be detected. Thus, the combination of SEIRAS and SERS investigations conclusively shows that only Cu sites on which CO adsorbs in the atop configuration while visible to SEIRAS are responsible for the CORR activity.

## Discussion

In summary, CO adsorption on Pt, Pd, Au, and OD-Cu surfaces was investigated in a broad range of electrochemical potentials with in situ surface-enhanced IR and Raman spectroscopies. We demonstrate that the two techniques generally probe different subpopulations of adsorbed CO on metal surfaces, e.g., CO adsorbed on terrace versus step sites on different facets, under identical conditions. The peak position and the Stark tuning rate of CO bands are largely similar on surfaces that bind CO more strongly, such as Pd and Pt, while substantial differences are observed on weakly CO-binding metals, e.g., Au and OD-Cu. Computational investigations suggest that these differences likely originate from the change in the distribution of CO based on the competitive adsorption of water. Combined spectroscopic and reactivity investigations show that only Cu sites corresponding to CO$_{atop}$ band visible in SEIRAS are active in the CORR, while the rest of the Cu sites are either unstable or unable to convert adsorbed CO.

## Methods

**Polishing and cleaning of Si crystal**. Before depositing metal films, Si prisms were thoroughly polished and cleaned. First of all, the Si prisms were immersed in fresh aqua regia solution, which was made with 75% HCl (Fisher Chemical) and 25% HNO$_3$ (Fisher Chemical), to remove any residual metal species on the surface. Then they were rinsed with deionized water (DI water) and dried with blowing air. Afterward, the Si crystals were polished with a slurry of 0.05 μm Al$_2$O$_3$ (Sigma-Aldrich) for several minutes until the surface being hydrophobic. Following that, the Si crystal was sonicated in alternate baths of DI water and acetone three times, to remove alumina powder and any organic residues on the surface. In the end, the Si crystals were dried with blowing air.

**Preparation of Pt films on Si crystals**. Polycrystalline Pt films were deposited on Si crystals by an electroless chemical plating method[37,38]. First of all, a Pd seed layer was directly deposited on the reflecting plane of the Si prism in order to improve the adhesion of Pt film[82]. The Pd deposition solution containing 0.23 mM PdCl$_2$ (99.999%, Alfa Aesar), 0.014 M HCl, 0.28 M HF (48%, 99.99% metals basis, Sigma-Aldrich), and 0.76 M NH$_4$F (40%, Sigma-Aldrich) was prepared in an aqueous solution. Before depositing the Pd seed layer, the Si crystal was first immersed in 40% NH$_4$F for 1 min and 45 s to make a hydrogen-terminated surface. Then it was immersed in the Pd plating solution at 50 °C for 3 min to obtain the Pd seed layer, after which it was rinsed using DI water, dried with blowing air, and sintered at 200 °C for 30 min in a vacuum evacuated tube furnace. The Pt plating solution consists two separate parts: (1) 0.01 M H$_2$PtCl$_6$·6H$_2$O (Sigma-Aldrich) aqueous solution; (2) 0.3 M NH$_3$ (30%, Fisher Scientific), 0.036 M HONH$_3$Cl (99.999% Aldrich), and 0.04 M N$_2$H$_4$·H$_2$O (98%, Sigma-Aldrich). These two solutions were mixed by the volume ratio of 1:1 immediately before Pt deposition. Then the Pd-seeded Si prism was immersed into the mixed Pt plating solution at 60 °C for several minutes, during which a voltmeter was used to check the conductivity of the film until a conductive Pt film was achieved. Then the obtained Pt film was rinsed with DI water and dried with blowing air.

**Preparation of Au film on Si crystal**. Polycrystalline Au film was deposited onto the Si crystal by chemical bath deposition[11]. Briefly, the polished Si crystal was first immersed in an NH$_4$F bath for 120 s to create a hydrogen-terminated surface. The Au plating solution consists of 5.75 mM NaAuCl$_4$·2H$_2$O, 0.025 M NH$_4$Cl, 0.025 M Na$_2$S$_2$O$_3$·5H$_2$O (98%), 0.075 M Na$_2$SO$_3$ (98%), and 0.026 M NaOH (99.99%). All chemicals were purchased from Sigma-Aldrich without further treatment. Before deposition, 0.8 mL of HF aqueous solution (diluted to 2 wt%) was mixed with 4.4 mL above Au plating solution. Then the hydrogen-terminated Si surface was immersed in the above mixture solution for 10 min at 55 °C. After the deposition, the Au film was rinsed using DI water and dried with blowing air.

**Preparation of Pd/Au film on Si crystal**. Polycrystalline Pd film was electro-deposited onto the obtained Au film on Si crystal[43]. The Pd plating solution consists of 5 mM PdCl$_2$ and 0.1 M HClO$_4$ (99.999%, Sigma-Aldrich). The electrodeposition was conducted at −200 μA until 100 mC of charge passed through the system, using an Au/Si crystal as the working electrode, a graphite rod as a counter electrode, and a saturated Ag/AgCl (BASI) as the reference electrode. Then the obtained Pd film was rinsed using DI water and dried with blowing air.

**Preparation of oxide-derived Cu (OD-Cu)/Au film on Si crystal**. OD-Cu film was prepared by electrodepositing Cu$_2$O onto the obtained Au film on Si crystal and the following electroreduction[14]. The electrolytic bath consists of 0.4 M CuSO$_4$ (99.99%, Sigma-Aldrich) and 3 M L-lactic acid (≥85%, Sigma-Aldrich). After dissolving the reagents, the pH of this electrolytic bath was adjusted to 11.5 by adding NaOH (99.99%, Sigma-Aldrich) at a constant temperature of 60 °C in a heated water bath. The electrodeposition was carried out potentiostatically under −0.4 V vs. Ag/AgCl using the typical three-electrode system as that in Pd plating process. During the electrodeposition, the water bath was kept under 60 °C and the deposition charge amount was controlled at 100 mC onto the gold substrate layer.

Afterward, the $Cu_2O$ electrode was rinsed using DI water and dried with blowing air. Before the spectroscopic tests, the $Cu_2O$ electrode was pre-reduced through in situ electrochemical reductions in 0.1 M $KHCO_3$ (prepared by purging $CO_2$ into $K_2CO_3$ (99.995%, Sigma-Aldrich) overnight until the pH reached 7.2) under a constant current density of $-500\ \mu A$ until more than 100 mC charge passed and the current density became stable. The obtained OD-Cu film was then rinsed using DI water and dried with blowing air.

**Preparation of Au@$SiO_2$ nanoparticles.** The Au@$SiO_2$ nanoparticles (NPs) were prepared following the method described below[31]. Briefly, Au NPs with an average size of 55 nm were prepared by adding 1.4 mL sodium citrate aqueous solution (1 wt%, 99%, Alfa Aesar) into 200 mL boiling $HAuCl_4$ aqueous solution (0.01 wt%, 99.99%, Sigma-Aldrich) under vigorous stirring. After that, the mixture was refluxed for 1 h and then cooled to room temperature. Following that, 0.6 mL 1 mM (3-Aminopropyl)triethoxysilane (APTES, 98%, Sigma-Aldrich) solution (pH 11) was added into 30 mL of Au NPs suspension solution and stirred for 15 min at room temperature. Then 3.2 mL sodium silicate solution (0.54 wt%, Sigma-Aldrich) was added to the above mixture and stirred for 5 min. After that, the mixture was kept in a 95 °C oil bath and stirred for another 30 min. The hot solution was then cooled in an ice bath followed by centrifugation at 3400×$g$ and washed with DI water. Finally, the concentrated Au@$SiO_2$ NPs were dispersed in 500 $\mu L$ $H_2O$.

**Materials characterization.** Scanning electron microscopy (SEM) images were obtained on a field emission scanning electron microscope (ZEISS Auriga 60 SEM/FIB). X-ray diffraction (XRD) spectra were collected on a Bruker D8 Discover diffractometer using a Cu Kα X-ray tube.

**Preparation of electrolytes.** The electrolytes used in this work were all pre-electrolyzed for 24 h at a constant reducing current of $-10$ mA in a Nafion membrane-separated (IEM, Nafion 211, Fuel Cell Store) two-compartment cell. The Cu foil (99.998%, Sigma-Aldrich) and a graphite rod were employed as working and counter electrodes, respectively. The pre-electrolyzing can deposit most of the metal impurities in the electrolytes onto the Cu foil.

**In situ SEIRAS tests.** In situ SEIRAS tests were conducted in a home-designed spectro-electrochemical cell with a three-electrode configuration as shown in Fig. 1a. The obtained metal film deposited on Si ATR crystal was used as the working electrode, a graphite rod as the counter electrode, and a saturated Ag/AgCl (BASI) as the reference electrode. The graphite rod was used as the counter electrode in order to avoid any metal contamination[83]. During the test, CO gas was kept bubbling into the electrolyte and the system was mechanically stirred. The potential on the cell was supplied by a Solartron 1260/1287 system for electrochemical measurements. SEIRA spectra were collected by an Agilent Technologies Cary 660 FTIR spectrometer equipped with a liquid nitrogen-cooled MCT detector. All spectra were collected at a $4\ cm^{-1}$ spectral resolution and presented in absorbance units where a positive and negative peak signifies an increase and decrease in the interfacial species, respectively. All SEIRA spectra presented in this work correspond to 64 coadded scans lasting about 40 s. After the SEIRAS experiment, the working electrode was removed from the SEIRAS cell, rinsed with DI water, and dried with blowing air.

**In situ SERS tests.** In situ SERS tests were conducted in a custom-designed three-electrode SERS flow cell as shown in Fig. 1b. In this setup, the electrolyte layer between the monochromatic laser and film surface is as thin as 5 mm to avoid the attenuation of scattering light. This flow cell also has two compartments that are separated by a piece of Nafion ion exchange membrane. Before the SERS test, 2 μL of Au@$SiO_2$ suspension was drop-casted onto the SEIRAS-used films when necessary to enhance the Raman signal. Then the Si crystal was fitted into the SERS flow cell and used as a working electrode, with a graphite rod as the counter electrode in an anodic cell, and a saturated Ag/AgCl (Thomas Scientific) as the reference electrode. SERS tests were performed on a LabRAM HR Evolution microscope (Horiba Jobin Yvon) equipped with a 632.8 nm He-Ne laser, a 50X objective (NA = 0.55), and a monochromator (600 grooves/mm grating), and a CCD detector. The scanning range is 100–2200 $cm^{-1}$ for each SER spectrum, which contains three grating windows, and the acquisition time for each grating window is set to 10 s. Each of the SER spectra presented in this work corresponds to two coadded scans and costs about 80 s. During the test, the fresh electrolyte pre-saturated with CO gas was kept flowing across the cell using an HPLC pump, which can not only replenish CO but also remove $H_2$ bubbles produced by the hydrogen evolution reaction (HER), thus avoiding the block of scattering light[8,9].

**Reactivity tests for the electrochemical CO reduction reaction (CORR).** The electrochemical reduction of CO on OD-Cu in this work was conducted in the SEIRAS spectrochemical cell shown in Fig. 1a. The OD-Cu film on a copper foil substrate was used as the working electrode, and a graphite rod and a saturated Ag/AgCl were used as the counter and reference electrodes, respectively. Before the reaction, the electrolyte in the cathode compartment was first purged with Ar,

during which the pre-reduction of $Cu_2O$ to OD-Cu was conducted under $-0.4$ V vs. RHE. The feeding gas was then switched to CO and continuously bubbled into the electrolyte for 0.5 h to reach saturation. Then the cell was sealed, and the reaction was conducted for 10 C of charge under $-0.7$ V vs. RHE in 0.05 M $K_2CO_3$ and 0.1 M KOH as shown in Supplementary Fig. 13. After the reaction, the gas products were detected by a GC (Agilent Technologies 7890B) equipped with a Flame Ionization Detector (FID) and Temperature Conductivity Detector (TCD). The liquid products were analyzed by NMR spectroscopy (Bruker AVIII 600) through an integrated peak area ratio with a 4 ppm DMSO/$D_2O$ internal standard.

**Computational methods.** Density functional theory (DFT) calculations were performed using the GPAW code along with the atomic simulation environment (ASE)[84–86]. The grid spacing for the real-space grid was set to 0.2 Å and Fermi smearing to 0.1 eV. All calculations were sampled with Monkhorst–Pack $k$-point grids of (4,4,1)[87]. A dipole correction was used in the direction parallel to the surface normal[88]. Continuum charge calculations were done with the solvated jellium model (SJM) implemented within GPAW[89]. The charge was added to the system in increments of 0.2 e from $-1$ e to $+1$ e. A dielectric constant of 78.36 was used. Relaxations were carried out using the Quasi-Newton algorithm implemented within ASE and geometries were converged to forces on all atoms <0.03 eV/Å[86]. Three surface facets, namely (111), (211), and (310), were investigated for the four metals of Au, Cu, Pd, and Pt. In all cases, we investigated slabs with three metal layers. $(3\times3)$ surface atoms were used for (111) and (211) and $(2\times3)$ for (310). The bottom two layers were kept fixed to mimic the bulk metal. On all metals, we considered CO adsorbed on three sites for (111), two sites for (211), and two sites for (310) as shown in Supplementary Fig. 12.

IR and Raman intensities were calculated based on the methodology used in the literature as follows[87,90].

1. Vibrational frequencies were computed using the ASE Vibrations class. Eigenmodes ($\omega$) were determined based on the calculated Hessian matrix. A displacement of atoms of 0.01 Å was used throughout.
2. To determine infrared intensities, $I_{IR} = \left|\frac{d\mu}{dQ}\right|$, where $\mu$ is the dipole moment and $Q$ is the normal mode (here, for the CO-stretch mode). We computed: (a) the derivative of $\mu$ with respect to small displacement $R$ in the $x$, $y$, $z$ Cartesian directions as a finite difference, $\frac{d\mu}{dR} = \frac{\mu_i - \mu_{-i}}{2\delta R}$. (b) The dipole derivative along the normal mode $\omega$ was calculated as $\frac{d\mu}{dQ} = \frac{d\mu}{dR} \cdot \omega$.
3. To compute Raman intensities, $I_{Raman} = \frac{d\alpha}{dR} \cdot \omega$, where $\alpha$ is the polarizability tensor: (a) energies were converged to $10^{-7}$ eV for each SCF cycle in order to accurately model the polarizability change with displacement (a second derivative quantity). Similar to the literature[59,90], we assume that fields in directions perpendicular to the surface normal are small in magnitude so that we consider only the $zz$ component of the polarizability tensor, $\alpha_{zz}$. We computed this through the second derivative of the forces $F$ with respect to the applied field $\xi$ in the $z$-direction, $\frac{d\alpha_{zz}}{dR} = \frac{d^2F}{d\xi^2}$, where $\xi$ is applied in the form of a saw-tooth potential. In practice, this quantity was computed using a finite-difference stencil as $\frac{d^2F}{d\xi^2} = \frac{(F_i - 2F_0 - F_{-i})}{d\xi^2}$, where $d\xi = 0.1$ V/Å and $F_i$, $F_0$, and $F_{-i}$ are the forces corresponding to the different fields.

The adsorption energies of CO* and $H_2O^*$ were determined using the methodology detailed in the literature[70]. Briefly temperature-programmed desorption data from experiments were used to determine desorption energy, $E_d$, which can be fit to the following functional form:

$$E_d = E_0 - b\theta_{sat}\theta_{rel} - k_B Tln\left(\frac{\theta_{rel}\theta_{sat}}{1 - \theta_{rel}\theta_{sat}}\right)$$

where $E_0$ is the adsorption energy at dilute coverage, $\theta_{sat}$ is the saturation coverage that occurs during dosage, $\theta_{rel}$ is the relative coverage which decreases as the temperature is increased and $b$ is a linear adsorbate–adsorbate interaction parameter.

## Data availability

Source data are provided with this paper. All relevant data are available from the authors on reasonable request.

## Code availability

Scripts used to generate the computational figures can be found at https://github.com/sudarshanv01/ir-raman-co or in ref. [91].

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

## Acknowledgements

This work is supported by the Beijing National Laboratory for Molecular Sciences, and the National Key R&D Program of China (grant number 2021YFA1501003). X.C., Y.Z, N.J.O., and B.X. also acknowledge the support of the National Science Foundation CAREER Program (Award No. CBET-1651625). S.V. and K.C. acknowledge VSUSTAIN Center 9455. The authors acknowledge computational resources from PRACE (project ID: prpa85) and the Juelich Supercomputing Centre.

## Author contributions

X.C. and B.X. conceived the idea and designed experiments in this study. X.C. carried out the in situ SEIRAS, SERS, and electrocatalytic experiments. X.C. and B.X. analyzed the experimental results. S.V. and K.C. conducted the DFT calculations and analyzed the data. Y.Z. synthesized Au@SiO2 nanoparticles and collected SEM images. N.J.O. synthesized chemically deposited Pt films. X.C., S.V., K.C., and B.X. wrote the manuscript.

## Competing interests

The authors declare no competing interests.
