## [Peer Review File · Nature Communications]

Title: Understanding the Complementarities of Surface-Enhanced Infrared and Raman Spectroscopies in CO Adsorption and Electrochemical ReductionEditorial Note: This manuscript has been previously reviewed at another journal that is not operating a transparent peer review scheme. This document only contains reviewer comments and rebuttal letters for versions considered at *Nature Communications*.

REVIEWER COMMENTS

Reviewer #1 (Remarks to the Author):

I have reviewed an earlier version of paper and provided some comments on the overall manuscript. It seems that the authors have tried to address those points. However, I feel that some key issues might have been circumvented.

For example, the effect of alkali metal is not well addressed. As I mentioned, not all experiments were performed in the presence of alkali metal. In the case of Pd/Pt, the spectroscopic behaviour was investigated in HClO₄. How shall this be applied to 'electrochemical CO reduction', as they claimed in the title? Should not 'Understanding the behaviour of CO adsorption' or similar claim be more appropriate? I fully understood that the key point of this study is to study the CO adsorption behaviour, which I appreciate very much. However, the assignment of atop/bridge CO shall be corroborated by other experiments as I mentioned earlier, either low wavenumber region or isotopic experiment. The ~2000 region is rather tricky and you might have seen quite some mistakes in assignments of these peaks in the literature. This part is not very convincing.

Apart from the above two points, I feel that the manuscript is presented in an excellent way.

Reviewer #2 (Remarks to the Author):

I appreciate the authors for their great efforts in revising the manuscript. Most of my concerns have been well addressed, but some of them remains. The authors should further convince the reviewer before publishing of this manuscript can be recommended.

1. As mentioned by the authors in the response letter, Raman spectra at different positions have slight difference, but it does not affect the conclusion. The related discussions should be added into the manuscript, which can make the conclusion more persuasive.

2. I do not agree that "The distortion in the cyclic voltammetry (CV) curves of Figure S3c is likely caused by the reduction of the SiO_x species on the Si ATR crystal on which Pt was deposited". Chemically deposited Pt film on Si usually gives very clear and symmetric CV features of Pt (e.g., *Journal of Electroanalytical Chemistry* 587 (2006) 299–307). If the significant distortion and featureless curves are caused by contamination or air leakage in the systems, the results may have some problems.

3. Regarding the response to my previous Comment 9, I still have concerns on the conclusion. I appreciate that authors cited reference that supports the stronger adsorption of water on step than on terrace. However, *CO adsorption is also well-computed to be stronger on step sites than terrace (J. Am. Chem. Soc. 2013, 135, 16833–16836, J. Am. Chem. Soc. 2014, 136, 6978–6986).

I believe the relative binding strength of CO versus H₂O on different sites is more relevant to determine whether CO should be pushed from step to terrace. In addition, whether the binding strength of *CO decreases at higher potentials, while the adsorption of water increases? If yes, it would lead to the decreased *CO coverage on both step and terrace, instead of pushing *CO from one site to another.

4. Regarding the response to my previous Comment 10, I am still confused. In Figure 3a, from 0.3 to 0.6 V, the CO band intensities on Raman and IR both do not change too much (seems <20%), suggesting almost unchanged CO coverage on different sites at these potentials (assuming these two tools are probing CO adsorbed on different sites). In consideration of the significantly different Stark rate in this potential window, I am not convinced that the CO coverage change on different sites caused by the competitive adsorption water is the reason for different apparent Stark tuning rate observed by Raman and IR.

Reviewer #3 (Remarks to the Author):

I have analysed again the manuscript entitled “Understanding the Complementarities of Surface-Enhanced Infrared and Raman Spectroscopies in the Electrochemical CO Reduction”. The manuscript is now improved and the authors have answered well to all my comments. Therefore, I have no questions from my point of view.

Reviewer 1:

General Comments: “I have reviewed an earlier version of paper and provided some comments on the overall manuscript. It seems that the authors have tried to address those points. However, I feel that some key issues might have been circumvented.”

Response: We thank the reviewer for the valuable time and constructive comments and will address the comments raised below.

Comment 1: “For example, the effect of alkali metal is not well addressed. As I mentioned, not all experiments were performed in the presence of alkali metal. In the case of Pd/Pt, the spectroscopic behaviour was investigated in HClO₄. How shall this be applied to 'electrochemical CO reduction', as they claimed in the title? Should not 'Understanding the behaviour of CO adsorption' or similar claim be more appropriate?.”

Response: Pt and Pd are largely inactive in the electrochemical CO reduction due to the strong CO adsorption or poisoning, and therefore, the alkali metal cations' effect in the CORR cannot be investigated on these two metals (Faradaic efficiency of CORR products is negligible and the majority of charges passed go into the hydrogen evolution reaction). Since the employment of Pt and Pd in this work is for the purpose of comparing the spectral features of adsorbed CO with the two techniques, we revised the title of this manuscript to “Understanding the Complementarities of Surface-Enhanced Infrared and Raman Spectroscopies in CO Adsorption and Electrochemical Reduction”.

Comment 2: “I fully understood that the key point of this study is to study the CO adsorption behaviour, which I appreciate very much. However, the assignment of atop/bridge CO shall be corroborated by other experiments as I mentioned earlier, either low wavenumber region or isotopic experiment. The ~2000 region is rather tricky and you might have seen quite some mistakes in assignments of these peaks in the literature. This part is not very convincing. Apart from the above two points, I feel that the manuscript is presented in an excellent way.”

Response: We agree with the reviewer that there are many inconsistencies in assignments of CO adsorbed bands in the literature. However, the difference mainly stems from the proposed correlation between the wavenumber and specific adsorption sites, e.g., the terrace and step/defect sites (*ACS Catal.* **2018**, *9*, 474; *Surf. Sci.* **1984**, *138*, 75; *Surf. Sci. Rep.* **1992**, *16*, 53; *J. Chem. Phys.* **1994**, *101*, 9080; *J. Phys. Chem. C* **2017**, *121*, 12337), rather than the type of bonding configurations, i.e., atop and bridge. This is because the bridge-bonded CO on a given surface is typically significantly lower in wavenumber (by 100 - 300 cm⁻¹) than atop-bonded CO (*ACS Catal.* **2018**, *8*, 7507; *Nature*, **2020**, *577*, 509; *Surf. Sci.* **2016**, *646*, 210). This is also in agreement with the calculated frequencies (Figure 4a-d of the main text). Our assignments of atop/bridge CO on Pt (*J. Phys. Chem. C* **2009**, *113*, 10222; *J. Phys. Chem.* **1989**, *93*, 5341; *Surf. Sci.* **2003**, *527*, 198), Pd (*Surf. Sci. Rep.* **1983**, *3*, 107; *Phys. Chem. Chem. Phys.* **2008**, *10*, 3662; *Surf. Sci.* **2016**, *646*, 210), Cu (*ACS Catal.* **2018**, *8*, 7507; *ACS Catal.* **2019**, *9*, 6305) and Au (*J. Am. Chem. Soc.* **2017**, *139*, 3774; *Surf. Sci.* **2007**, *601*, 1898; *J. Electroanal. Chem.* **1982**, *142*, 345) are consistent with the literature.

The peak assignment of adsorbed CO on atop site of the Pt surface is consistent with the SER band of Pt-C stretching mode in the low wavenumber region (Figure R1a). The sharp peak at 520.7 cm⁻¹ lack of shift with the potential originates from the Si crystal on which the Pt film was deposited. Another band centered at ~480 cm⁻¹ is assigned to the Pt-C stretching mode of CO adsorbed on the top site of Pt, which is consistent with the literature (*J. Phys. Chem.* **1970**, *74*, 4335; *Angew. Chem. Int. Ed.* **2020**, *59*, 23554). This peak blueshifts from 481 to 486 cm⁻¹ with the cathodic sweep of potential from 0.6 to 0 V, resulting in a Stark tuning rate of ~8 cm⁻¹/V. This trend differs from that observed on C-O band which redshifts with the cathodic sweep of potential (Figure R1b). These observations are expected based on the fact that the degree of d-π* back-bonding between Pt and C enhanced with decreasing the electrode potential (*J. Phys. Chem. B* **1997**, *101*, 5842). In addition, it can be found that the Raman intensities of both the Pt-C and C-O bands tracks with each other in the whole potential range (Figure R1a and b), indicating a shared origin.

Figure R1. (a) The low wavenumber region and (b) the C-O band region of in-situ SER spectra on polycrystalline Pt film in CO saturated 0.1 M HClO₄ (pH 1.2). Figure R1b is a copy of Figure S2b.

In addition, we conducted the isotopic experiments of ¹³CO and ¹²CO adsorption on OD-Cu by SEIRAS to further confirm that these two bands are CO-related. Initially, the spectra were collected under ¹³CO (1.0 atm) in 0.1 M KOH (pH = 12.9) at -0.3 V_{RHE}, at which no appreciable CORR occurred. The bands at 2015 and 1781 cm⁻¹ are attributed to ¹³CO_{ad} at atop and bridge sites on Cu, respectively, at this potential (orange trace in Figure R2). After reaching the adsorption equilibrium of ¹³CO, ¹²CO was introduced into the electrolyte and the SEIRA spectra were collected continuously for 10 min (green and black traces in Figure R2). The ¹³CO_{ad} bands gradually decrease in intensity with the rise of ¹²CO_{ad} bands, which blueshift and locate at 2062 and 1826 cm⁻¹ for ¹²CO_{atop} and ¹²CO_{bridge}, respectively, after reaching the adsorption equilibrium (top green trace in Figure R2). Both of the CO_{atop} and CO_{bridge} bands blueshift ~46 cm⁻¹ after the substitution of ¹³CO by ¹²CO, which is consistent with previous reports (*J. Am. Chem. Soc.* **2017**, 139, 3774; *Surf. Sci.* **1979**, 89, 486; *Surf. Sci.* **1977**, 68, 528) and also the theoretical results based on the following equation:

$$\frac{1}{\lambda} \sqrt{\frac{\mu^*}{\mu}} = \frac{1}{\lambda^*}$$

where $1/\lambda$ and $1/\lambda^*$, and μ and μ^* are the wavenumber of vibrational modes and reduced masses of the unlabeled and labeled molecules, respectively. Therefore, the bands centered at ~2060 cm⁻¹ and ~1850 cm⁻¹ regions are reasonably assigned to CO adsorbed on atop and bridge sites of Cu, respectively.

Figure R2. (a) In-situ SEIRAS study on the substitution of ¹³CO_{ad} by ¹²CO_{ad} on OD-Cu at -0.3 V_{RHE} in 0.1 M KOH (pH = 12.7). Data in this figure has been used in another manuscript, so this figure is for review purpose only.

Action: We included Figure R1a in the supporting information and added the following sentence and the citation

in the main text:

“Complementary Pt-C stretching mode was also observed in SER spectrum at lower wavenumbers with expected Stark shifts (Figure S3)⁴⁵.” (Line 4, Page 9)

Reviewer 2:

General Comments: “I appreciate the authors for their great efforts in revising the manuscript. Most of my concerns have been well addressed, but some of them remains. The authors should further convince the reviewer before publishing of this manuscript can be recommended.”

Response: We thank the reviewer for the constructive comments and will address the comments raised below.

Comment 1: “As mentioned by the authors in the response letter, Raman spectra at different positions have slight difference, but it does not affect the conclusion. The related discussions should be added into the manuscript, which can make the conclusion more persuasive.”

Response: We thank the reviewer for pointing this out. We have added the related discussions in the main text to make the conclusion more persuasive.

Action: We added the following sentences in Line 21, Pages 12 in the main text:

“It is worth noting that Raman spectra collected at different spots of the electrode may vary slightly as we reported on Cu microparticles recently⁹. However, variations in peak position among different spots are typically within 10 cm^{-1} and cannot account for the substantial differences (up to 47 cm^{-1}) between IR and Raman spectra (Figure 3).”

Comment 2: “I do not agree that “The distortion in the cyclic voltammetry (CV) curves of Figure S3c is likely caused by the reduction of the SiO_x species on the Si ATR crystal on which Pt was deposited”. Chemically deposited Pt film on Si usually gives very clear and symmetric CV features of Pt (e.g., *Journal of Electroanalytical Chemistry* 587 (2006) 299–307). If the significant distortion and featureless curves are caused by contamination or air leakage in the systems, the results may have some problems.”

Response: The quality of CV on Pt films deposited on Si crystals depends heavily on the thickness of the Pt film. It can be clearly observed by the SEM image in Figure S1b that Si substrate surface is indeed exposed to the electrolyte due to the large particle size and heterogeneity of chemically deposited Pt. The surface SiO_x species are known to reduce at negative potentials, which could contribute to the generally downward slant in the CVs (Figure S4c). In order to further confirm the interference of Si substrate to the CVs, we deposited Pt film onto a gold substrate layer on Si crystal through electrodeposition, which can form a relatively uniform film as shown in Figure R3a. Although there still exists some holes on the film, the CVs on this Pt film (Figure R3b) are much better than those measured on chemically deposited one (Figure S4c). The similar CV curves obtained in SEIRAS (blue) and SERS (green) cells indicate that the difference in the cell configuration for these two spectroscopies does not impact on the reactivity in any substantial way, which is the main purpose of conducting CV tests by these two different cells. The SEIRA spectra of CO on this film (Figure R3c) are similar to those presented in the manuscript (Figure S2a and c), suggesting that although the underlying SiO_x species affects the quality of CV traces, it does not have a substantial impact on spectroscopic features. We thank the reviewer for pointing to the beautiful work by Kunimatsu *et al.* (*J. Electroanal. Chem.* **2006**, 587 299), where they deposited Pt films with CV features comparable to that on polycrystalline Pt surface in RDE studies.

Figure R3. (a) SEM image of Pt film deposited on gold substrate layer. (b) Cyclic voltammetry curves on the Pt film in (a) in Ar sat. 0.1 M HClO₄ using SEIRAS (blue) and SERS (green) cells, respectively. (c) In-situ SEIRAS spectra on the Pt film electrochemically deposited on a Au substrate in CO saturated 0.1 M HClO₄ (pH 1.2).

Comment 3: “Regarding the response to my previous Comment 9, I still have concerns on the conclusion. I appreciate that authors cited reference that supports the stronger adsorption of water on step than on terrace. However, *CO adsorption is also well-computed to be stronger on step sites than terrace (*J. Am. Chem. Soc.* 2013, 135, 16833–16836, *J. Am. Chem. Soc.* 2014, 136, 6978–6986).

*I believe the relative binding strength of CO versus H₂O on different sites is more relevant to determine whether CO should be pushed from step to terrace. In addition, whether the binding strength of *CO decreases at higher potentials, while the adsorption of water increases? If yes, it would lead to the decreased *CO coverage on both step and terrace, instead of pushing *CO from one site to another.”*

Response: We thank the reviewer for this comment and raising two important references. We agree that CO* binding strengths are increased as the coordination number of the metal is decreased, so CO binds stronger on a step than a terrace site under vacuum conditions. We added the citation of the literature that the reviewer mentioned.

Action: We added the citation of the above two references in Line 13, Pages 17 in the main text:

“The higher the coordination number of adsorbed CO has, the weaker the C-O bond is^{64,65}, leading to a lower wavenumber for the C-O stretching mode.”

We have shown that CO is more likely to bind to the (100) terrace than (211) steps in an electrochemical environment as reported by our recent work (*J. Phys. Chem. C* **2021**, 125, 17684). We agree with the reviewer that it is the relative binding strength between CO and H₂O that matters in determining the adsorption sites and surface coverage. Figure 5a illustrates the effect that the relative binding strength between these two species has on the coverage of CO_{ad}. If the CO_{ad} binding strength is stronger than that of H₂O_{ad}, the coverage of CO_{ad} on that site will be high (red region). In contrast, in the case of weaker binding strength of CO_{ad} than that of H₂O_{ad}, or both bind weakly, CO_{ad} coverage will be low (blue region). We emphasized this point in the main text.

Action: We added the following sentences in Line 1, Pages 20 in the main text:

“Thus, the coverage of adsorbed CO at a particular site is determined by the relative binding energy of CO and H₂O, i.e., the stronger the CO binding strength against that of H₂O, the more likely that there would be a higher CO* coverage on that site (red region in Figure 5a).”

To clarify, we do not suggest that the binding strength of CO* changes significantly with potential. The only source of change in ΔG_{CO} might come from an increase in interaction between CO* and the interfacial field, which is predicted to be about 0.1 eV over 0.5 V range based on our previous DFT calculation results (*J. Phys. Chem. C* **2021**, *125*, 17684; *J. Am. Chem. Soc.* **2017**, *139*, 11277). Given that the CO* binding strength does not change significantly with potential, while the water adsorption does change with potential, we would expect step sites of weak binding metals to have a coverage of CO* at potentials where water binding is weakened and vice versa. We clarified this point clearer in the revised text.

Action: We added the following sentences in Line 9, Pages 20 in the main text:

“Meanwhile, we do not expect significant change in CO binding strength with potential. Previous DFT calculations in the presence of an interfacial field suggest a change in CO binding strength of about 0.1 eV over 0.5 V range^{69,84}.”

The change in CO* site with potential means that the C-O stretch frequency changes with potential as well, as shown in the schematic of Figure 5b.

Comment 4: “Regarding the response to my previous Comment 10, I am still confused. In Figure 3a, from 0.3 to 0.6 V, the CO band intensities on Raman and IR both do not change too much (seems <20%), suggesting almost unchanged CO coverage on different sites at these potentials (assuming these two tools are probing CO adsorbed on different sites). In consideration of the significantly different Stark rate in this potential window, I am not convinced that the CO coverage change on different sites caused by the competitive adsorption water is the reason for different apparent Stark tuning rate observed by Raman and IR.”

Response: We thank the reviewer for this comment. We did not present all the spectra under each potential in Figure 3a to streamline the flow of the manuscript. All CO_{ad} peaks presented in Figure 3a have been integrated above 60% of the maximum values in both SEIRAS and SERS to allow for reliable determinations of Stark tuning rates (to remove the effect of the dynamic dipole coupling on the peak position) as reported in our recent work (*Catal. Sci. Technol.* **2021**, *11*, 6825). All the spectra and the integrated CO_{ad} peak area in the potential range of -0.2 to 0.8 V are included in Figures S7 and S8 (presented below as Figure R4). It can be found that the variations of peak intensity are much more significant in SEIRA (Figure R4b) than in SER (Figure R4d) spectra in the same potential range of 0.3 to 0.8 V. We do not argue that the change in the overall CO coverage leads to the different apparent Stark tuning rates in Raman and IR spectra, but hypothesize that the shift of CO adsorbed on step sites to terrace sites as the potential decreases (from 0.8 to 0.3 V) is the cause. As the potential decreases on the Au surface from 0.8 to 0.3 VRHE (the determined potential of zero charge on Au is 0.4 VSHE, i.e., 1.0 VRHE in 0.1 M KHCO₃), water adsorption becomes more favorable due to its high polarity, which would give rise to a decreasing CO coverage on steps.

Water adsorption strength will increase with the potential (*Sci. Adv.* **2020**, *6*, eabb1219), and the CO adsorption energy is less sensitive to potential, as reported in our previous work (*J. Phys. Chem. C* **2021**, *125*, 17684). Thus, a fraction of CO adsorbed at step sites will likely migrate to terrace sites as potential increases due to the more effective water adsorption. Since CO adsorbed on terrace sites exhibits a higher wavenumber than step-bonded CO, the potential induced shift in adsorption sites is expected to enhance the apparent Stark tuning rate. This effect is more pronounced in the infrared than Raman spectra because the calculated changes in the dipole moment derivative in adsorbed CO is more sensitive to a shift from the highly stepped Au(310) facet to Au(111) than that of the polarizability derivative (Figure 4f). We acknowledge that this is one possible explanation of the observed difference in the apparent Stark tuning rates with these two techniques, and further studies are needed to confirm this hypothesis.

Figure R4. (a) In-situ SEIRA spectra on polycrystalline Au film in CO saturated 0.1 M KHCO_3 (pH 8.9). The background was collected at -0.4 V under Ar purge. (b) The normalized peak area of the CO adsorption bands as a function of potential in the range of 0.7 to 0.3 V. (c) The potential dependence of CO band frequency. Stark tuning rate is determined through the linear fits of the point data between 0.7 and 0.3 V, at which the peak area is greater than 60% of the maximum. (d) Tandem in-situ SER spectra on the same Au film under identical conditions as those in Figure R4a. (e) The potential dependence of CO band frequency. Stark tuning rate is determined through the linear fits of the point data between 0.8 and 0.2 V, at which the peak area is greater than 60% of the maximum.

In order to make this point clearer, we added and revised the following sentences and added some citations in the main text:

Action: We added and revised the following sentences in Line 13, Pages 20 in the main text:

“The computed Stark tuning rates (Figure 4a) are approximately $20 \text{ cm}^{-1}/\text{V}$ on all sites. Figure 5b (right panel) shows schematically a possible mechanism through which the electrode potential could impact on the apparent Stark tuning rate based on our hypothesis. As the potential increases on the Au surface from 0.3 to $0.8 V_{\text{RHE}}$, water adsorption becomes more favorable^{82,83,85}, which would give rise to a decreasing CO coverage on steps⁶⁹. The large apparent Stark tuning rate would result from the gradual shift in the predominance of the signal of adsorbed CO on steps at potentials at lower potentials to that of adsorbed CO on terraces at higher potentials (see schematic in Figure 5b’s left panel), as CO adsorbed on terrace sites have a higher wavenumber than those on step sites based on the computational results (Figure 4a). This effect is more pronounced in the infrared than Raman spectra because the calculated changes in the dipole moment in adsorbed CO is more sensitive to a shift from the highly stepped Au(310) facet to Au(111) than that of the polarizability (Figure 4f), which qualitatively explains the discrepancy of the measured Stark tuning rates with the two techniques.”

REVIEWER COMMENTS

Reviewer #1 (Remarks to the Author):

The manuscript is now in a good shape, thus being suitable for publishing as it is.

Reviewer #2 (Remarks to the Author):

The authors have well addressed most of my concerns in this version. As I mentioned in previous two versions, the spectra data presented are of high quality, and this is a nice manuscript. But I still have some suggestions that may help further polish the manuscript.

1. I would like to thank the authors for bringing up more details to their JPCC 2021 paper. Now I understand that the relative binding strength of CO on step and terrace may be different in vacuum and with water layer. As mentioned by the authors, the water binding increases at higher potentials, while the binding of CO should not change too much. Now I am convinced that as the potentials increases, the CO coverage decreases. But I suppose that occurs on both terrace and step sites, but not on the step only. Thus, it is problematic with authors' argument in the Response Letter that "Thus, a fraction of CO adsorbed at step sites will likely migrate to terrace sites as potential increases due to the more effective water adsorption". If this occurs, the CO coverage on the terrace will increase.

2. As mentioned by the authors "The large apparent Stark tuning rate would result from the gradual shift in the predominance of the signal of adsorbed CO on steps at lower potentials to that of adsorbed CO on terraces at higher potentials (see schematic in Figure 5b's left panel), as CO adsorbed on terrace sites have a higher wavenumber than those on step sites based on the computational results (Figure 4a)." I would like to mention that even without the "migration of CO from step to terrace mechanism", the hypothesis may still be valid. As long as the CO coverage decrease is more obvious on step than on terrace sites when the potential increases, the IR signal will show more features from terrace-bonded CO than step-bonded one. Hence, the apparent Stark tuning rate can still be larger than Raman.

3. At last, I would like to suggest the authors to modify Figure 5b (right) to avoid confusion, if they agree that CO coverage on terrace and step sites should both decrease as the potential increases.

Below we provide a point-by-point response to reviewers' comments along with the relevant changes in the revised manuscript.

Reviewer 2:

General Comments: “The authors have well addressed most of my concerns in this version. As I mentioned in previous two versions, the spectra data presented are of high quality, and this is a nice manuscript. But I still have some suggestions that may help further polish the manuscript.”

Response: We thank the reviewer for the valuable time and constructive suggestions and will address the comments raised below.

Table R1. ΔE of CO* adsorption on Au(211), Cu(211) stepped and Au(100), Cu(111) terrace surfaces

System	ΔE in vacuum	ΔE in presence of water	Ref.
Au(211)-stepped surface	-0.47 eV	$0.28 \text{ eV} \pm 0.14 \text{ eV}$	J. Phys. Chem. C 2021 , 125, 17684
Au(100)-terrace surface	-0.35 eV	$-0.52 \text{ eV} \pm 0.31 \text{ eV}$	
Cu(211)-stepped surface	-0.93 eV	$-0.69 \text{ eV} \pm 0.05 \text{ eV}$	J. Chem. Phys. 2020 , 152, 144703
Cu(111)-terrace surface	-0.78 eV	$-0.77 \text{ eV} \pm 0.06 \text{ eV}$	

Comment 1: “I would like to thank the authors for bringing up more details to their JPCC 2021 paper. Now I understand that the relative binding strength of CO on step and terrace may be different in vacuum and with water layer. As mentioned by the authors, the water binding increases at higher potentials, while the binding of CO should not change too much. Now I am convinced that as the potentials increases, the CO coverage decreases. But I suppose that occurs on both terrace and step sites, but not on the step only. Thus, it is problematic with authors' argument in the Response Letter that “Thus, a fraction of CO adsorbed at step sites will likely migrate to terrace sites as potential increases due to the more effective water adsorption”. If this occurs, the CO coverage on the terrace will increase.”

Response: We thank the reviewer for this important comment. Our *ab-initio* molecular dynamics simulations on Au (summarized from Ref. *J. Phys. Chem. C* **2021**, 125, 17684. and *J. Chem. Phys.* **2020**, 152, 144703. in Table R1 for Au and Cu) suggest that terraces tend to bind CO* slightly stronger or unchangeably in the presence of water, while stepped surfaces do the opposite, i.e., bind CO* weaker. In electrochemical conditions, we suggest that the reaction intermediate is stabilized from solvation of the reaction intermediates, while it is destabilized by water adsorption. Since terraces tend to bind water weaker than steps, the solvation of CO* leads to stronger binding on terraces, but not for steps.

We note that our findings on Au and Cu might not be directly transferrable to other metals, i.e., the terrace sites of stronger binding metals such as Pt might be destabilized. We clarified this point in the text:

Action: We added the following sentences in the main text:

“As the potential increases on the Au surface from 0.3 to 0.8 V_{RHE} , water adsorption has been calculated to be more favorable^{82,83,85}, which would give rise to a decreasing CO coverage on steps⁶⁹. Meanwhile, the coverage of CO on terrace sites, which are less affected by water adsorption, is expected to either increase or remain constant based on the calculated adsorption energies of CO^{69,86}.” (Line 4, Page 21)

Comment 2: “As mentioned by the authors “The large apparent Stark tuning rate would result from the gradual

shift in the predominance of the signal of adsorbed CO on steps at lower potentials to that of adsorbed CO on terraces at higher potentials (see schematic in Figure 5b's left panel), as CO adsorbed on terrace sites have a higher wavenumber than those on step sites based on the computational results (Figure 4a).” I would like to mention that even without the “migration of CO from step to terrace mechanism”, the hypothesis may still be valid. As long as the CO coverage decrease is more obvious on step than on terrace sites when the potential increases, the IR signal will show more features from terrace-bonded CO than step-bonded one. Hence, the apparent Stark tuning rate can still be larger than Raman.”

Response: We thank the reviewer for this important comment. We agree with the reviewer that the hypothesis holds even in the case where CO* coverage decreases more steeply for steps than that on terraces, i.e., a migration from steps to terraces is not necessary for the larger Stark tuning rates. We added this possible explanation in the main text.

Action: We added the following sentences in the main text:

“The large apparent Stark tuning rate would result from the gradual shift in the predominance of the signal of adsorbed CO on steps at lower potentials to that of adsorbed CO on terraces at higher potentials (see schematic in Figure 5b's left panel), as CO adsorbed on terrace sites have a higher wavenumber than those on step sites based on the computational results (Figure 4a). We note that a larger apparent Stark tuning rate would also appear if the coverage of CO on steps and terraces decreases with the decrease more significant on stepped sites.” (Line 10, Page 21)

Comment 3: “At last, I would like to suggest the authors to modify Figure 5b (right) to avoid confusion, if they agree that CO coverage on terrace and step sites should both decrease as the potential increases.”

Response: We thank the reviewer for their attention to this detail. We agree that it is possible that the coverage of CO decreases in certain cases; though as seen in Table R1 in Comment 1, it does not for Au and Cu. In order to keep the schematic general, we have incorporated the reviewer's suggestion and showed that CO adsorbed on a step site can either go to the terrace or be removed.

Action: We revised Figure 5b in the main text:

Figure 5. a) Boltzmann coverages based on different ΔG_{CO} and ΔG_{H_2O} values with experimental data shown for Au(310)⁷⁵, Pt(111)^{76,77}, Ni(111)^{78,79} and Rh(111)^{80,81}. b) Schematics showing the effect of change in CO*

coverage on observed vibrational frequencies as a function of potential (left) and cartoon illustrating the effect of competitive water adsorption on CO* binding site (right).

REVIEWERS' COMMENTS

Reviewer #2 (Remarks to the Author):

The manuscript is ready for publish.